# Salicylic Acid-Producing Endophytic Bacteria Increase Nicotine Accumulation and Resistance against Wildfire Disease in Tobacco Plants

**DOI:** 10.3390/microorganisms8010031

**Published:** 2019-12-22

**Authors:** Md. Nurul Islam, Md. Sarafat Ali, Seong-Jin Choi, Youn-Il Park, Kwang-Hyun Baek

**Affiliations:** 1Soil Resource Development Institute, Regional Office, Rajshahi 6000, Bangladesh; nurulsrdi78@gmail.com; 2Department of Biotechnology, Yeungnam University, Gyeongsan, Gyeongbuk 38541, Korea; sarafatbiotech@ynu.ac.kr; 3Department of Biotechnology, Catholic University of Daegu, Gyeongsan 38430, Korea; sjchoi@cu.ac.kr; 4College of Biological Sciences and Biotechnology, Chungnam National University, Daejeon 34134, Korea; yipark@cnu.ac.kr

**Keywords:** *Pseudomonas tremae*, endophytic bacteria, salicylic acid, disease resistance, wildfire disease, nicotine synthesis

## Abstract

Endophytic bacteria (EB) are both a novel source of bioactive compounds that confer phytopathogen resistance and inducers of secondary metabolites in host plants. Twenty-seven EB isolated from various parts of *Metasequoia glyptostroboides, Ginkgo biloba, Taxus brevifolia, Pinus densiflora, Salix babylonica*, and *S. chaenomeloides* could produce salicylic acid (SA). The highest producers were isolates EB-44 and EB-47, identified as *Pseudomonas tremae* and *Curtobacterium herbarum*, respectively. *Nicotiana benthamiana* grown from EB-44-soaked seeds exhibited a 2.3-fold higher endogenous SA concentration and increased resistance against *P. syringae* pv. *tabaci*, the causative agent of tobacco wildfire disease, than plants grown from water-soaked seeds. *N benthamiana* and *N. tabacum* grown from EB-44-treated seeds developed 33% and 54% disease lesions, respectively, when infected with *P. syringae* pv. *tabaci*, and showed increased height and weight, in addition to 4.6 and 1.4-fold increases in nicotine accumulation, respectively. The results suggest that SA-producing EB-44 can successfully colonize *Nicotiana* spp., leading to increased endogenous SA production and resistance to tobacco wildfire disease. The newly isolated EB can offer an efficient and eco-friendly solution for controlling wildfire disease and nicotine accumulation in *Nicotiana*, with additional application for other important crops to increase both productivity and the generation of bioactive compounds.

## 1. Introduction

Endophytes comprise fungi or bacteria that colonize plant tissues without harming the host plant [1]. Bioactive compounds from endophytes, such as *Bacillus, Pseudomonas,* and *Rhizobium*, are known to promote host plant growth [2,3], whereas endophytic bacteria (EB) can confer resistance to pathogen-induced diseases in the host plant [4,5,6,7,8]. Bacterial species such as *P. fluorescens* (strain CHA0), *P. aeruginosa* (7NSK2), and *Serrtia macrcescens* (strain 90-166) produce salicylic acid (SA) and their colonization of host plants increases the endogenous SA levels, in addition to enhancing host defences [9,10,11]. The results of EB application from several studies have indicated that SA promotes plant growth and stimulates plant defence mechanisms by inducing systemic acquired resistance (SAR) [12,13,14].

Wildfire disease caused by *Pseudomonas syringae* pv. *tabaci* (*Pst*) is one of the most destructive tobacco diseases [15,16,17]. This disease is characterized by the emergence of small brown or black water-soaked lesions surrounded by a broad, chlorotic halo on leaves, resulting from exotoxin production by *Pst* [18]. An integrated approach for the management of wildfire disease consists of (i) using seeds treated with silver nitrate; (ii) using certified tobacco seeds, rotating seedbed sites, and properly fumigating seedbed areas; (iii) spraying with a suitable copper-based compound and acibenzolar-S-methyl; (iv) spraying with antibiotics; and (v) using wildfire-resistant cultivars [18,19]. Application of copper-based compounds in combination with acibenzolar-S-methyl and streptomycin is especially effective at controlling the disease [18]. Since 1955, streptomycin has been used as a major antibiotic to control various bacterial diseases, including fire blight, soft rot, bacterial spot, and wildfire diseases of different crops [20]. However, streptomycin-resistant bacterial strains have emerged because of indiscriminate and long-term use of this antibiotic [21,22]. A recent study showed that tobacco wildfire disease infection rates were significantly decreased by the foliar application of a bacterial mix of *Bacillus* (87.74%), *Alcaligenes* (7.69%), *Pseudochrobactrum* (2.86%), and *Achromobacter* (1.05%) [23]. Although development of wildfire-resistant cultivars may be an alternative and eco-friendly method of controlling the disease, this process is time-consuming and laborious and these cultivars are not widely available. Therefore, development of a practical and convenient alternative for controlling wildfire disease is urgently required.

Nicotine, a plant alkaloid, is the most important compound in tobacco [24]. Nicotine plays important roles in tobacco plants, including in protection against insects and herbivores and regulation of plant growth [25,26]. Inoculation with phytopathogenic *Pst* and *P. syringae* pv. *tomato* results in increased nicotine levels in tobacco plants [27,28,29], while multiple signal molecules, including jasmonic acid (JA) and auxin, can also stimulate the synthesis of nicotine [30]. Numerous studies have reported the isolation of nicotine-degrading bacteria, including *Pseudomonas* sp. [31,32], *Ochrobactrum intermedium* [33], *Rhodococcus* sp. [34], and *Agrobacterium* sp. S33 [35]. To the best of our knowledge, however, there are no reports indicating that EB treatment promotes nicotine accumulation in tobacco.

In this study, we isolated EB from different tissues of six plant species and measured SA production. Endophytic bacteria that constitutively produce SA could provide a means to develop SAR against wildfire disease in tobacco through successful colonization of the host tobacco plants. To investigate the potential beneficial effects of SA-producing EB on host plants, we applied SA-producing EB to *Nicotiana benthamiana* and *N. tabacum* L. and evaluated resistance against wildfire disease, plant growth, and nicotine accumulation.

## 2. Materials and Methods

### 2.1. Collection of Plant Materials and Isolation of EB

Leaves, fruits, seeds, and cones of gymnosperm and angiosperm species, including *Metasequoia glyptostroboides, Ginkgo biloba, Taxus brevifolia, Pinus densiflora, Salix babylonica*, and *S. chaenomeloides*, were collected from Yeungnam University campus, Gyeongsan, Republic of Korea, in October 2014. The collected plant material was immediately taken to the laboratory in plastic bags for EB isolation. Plant tissues with visible superficial injury were excluded, and only healthy tissues were used for EB isolation.

Isolation of EB was carried out as per the standard procedure [36]. To isolate single EB colonies, the ground tissue extracts were diluted 1 × 10^−1^ and 10^−2^-fold with a sterilized NaCl solution (0.9%), spread onto plates containing four different types of media (25% yeast extract, nutrient broth, and agar (YNA), 25% nutrient agar (NA), de Man, Rogosa, and Sharpe (MRS), and Luria–Bertani (LB)), and incubated for up to 15 days at 28 °C. All the colonies were counted and the values were expressed as colony-forming units (CFUs) per gram of fresh tissue. This single colony isolation procedure was repeated three times for confirmation of each isolated EB, and morphological characteristics such as the form, margin, colour, and height of the colonies were documented. Isolated EB were cultured in the original isolation media and stored as 50% glycerol stocks at −80 °C until use.

The isolated EB were identified based on 16S rRNA gene sequencing. Sequencing data, alignment, and the constructed phylogenetic tree were analysed using Molecular Evolutionary Genetics Analysis (MEGA) software [37,38].

### 2.2. Measurement of SA Production in Isolated EB and Host Plants

In vitro screening for SA-producing EB was carried out following the standard protocol [9,10], with slight modifications. For inoculation, 20 µL aliquots of the glycerol stocks of each EB culture were diluted in 5 mL of casamino acid broth (CAB) in 50 mL Falcon tubes and incubated at 28 °C for 36 h at 180 rpm in the dark. Subsequently, 100 µL of this culture was transferred to 30 mL of CAB, incubated for 36 h, and centrifuged at 3500 rpm for 15 min at 4 °C, and the supernatant was collected. The pH of the supernatant was adjusted to ~1.5–2.0 with 1 M HCl, following which the supernatant was mixed with an equal volume of ethyl acetate and incubated overnight at room temperature. The ethyl acetate layer was collected the following day and evaporated at 50 °C on a rotary evaporator (A-1000S, EYELA, Tokyo, Japan). The dried extracts were collected in tubular glass vials, suspended in 1 mL of HPLC-grade methanol, and then filtered through a 0.45-µm syringe filter (SN-01345, Thermo Fisher Scientific, Waltham, MA, USA).

Endogenous SA was quantified according to the method described by Press et al. [10], with minor modifications. The filtrated 20 µL solution of each EB was injected into an HPLC system consisting of a pump equipped with an Eclipse XDB-C18 column (5 µm, 4.6 × 250 mm, Agilent Technologies, Santa Clara, CA, USA) and a UV detector (YL9100, Young-Lin, Korea). The isocratic mobile phase containing 0.2 M sodium acetate buffer (pH 5.5) in 10% methanol was applied at a flow rate of 0.8 mL/min. The wavelength and column temperature were set at 302 nm and 40 °C, respectively. The levels of SA were quantified by comparing the area of the corresponding peaks with the standard curve created using different concentrations (500, 250, 125, 62.5, and 31.25 µg/mL) of free SA (Duchefa Biochemie, Haarlem, The Netherlands). Quantification of the SA produced by the EB was repeated three times using HPLC.

The concentrations of free SA in the fresh leaves of *M. glyptostroboides, G. biloba, T. brevifolia, Pinus densiflora*, *S. babylonica*, *S. chaenomeloides*, and *N. benthamiana* were measured, as described by Dhakal et al. [39], using approximately 0.5 g of leaves.

### 2.3. Growth of Tobacco Species Inoculated with SA-Producing EB

Among the 27 EB isolates that produced SA, the highest SA-producing strains, EB-44 and EB-47, and a non-SA-producing strain, EB-52, were grown in culture broth, centrifugated at 3500 rpm for 15 min at 4 °C, and resuspended to OD_600_ 0.5 in 10 mM MgCl_2_. As a positive control, 1 mM salicylic acid in 0.3% Tween 20 was sprayed on 7-week-old *N. tabacum* plants growing in a glasshouse.

Seeds of *N. benthamiana* and *N. tabacum* were soaked for 3 h in a suspension containing SA-producing EB at OD_600_ = 0.5 or 10 mM MgCl_2_. The seeds were then sown in plastic pots (12 cm diameter × 10 cm height) containing 17% peat moss, 70% coco peat, 5% zeolite, and 8% perlite and grown in a controlled walk-in chamber. Seedlings of *N. tabacum* were transferred to the glasshouse for further experiments and *N. benthamiana* seedlings were grown under fluorescent light at 120 μE/(m^2^·s) under a 16/8 h light/dark photoperiod at 23 °C in a walk-in chamber.

### 2.4. Disease Assay of Pst-Infected Tobacco Species

Disease assays were performed by infecting *N. benthamiana* and *N. tabacum* with the wildfire disease pathogen *Pst*. The bacteria were grown at 28 °C in King B medium supplemented with 50 mg/mL rifampicin. Overnight bacterial cultures were harvested by centrifugation at 3500 rpm for 15 min at 4 °C and then resuspended at OD_600_ 0.1 in 10 mM MgCl_2_. Leaves of 4-week-old *N. benthamiana* and 6-week-old *N. tabacum* plants were infiltrated with 0.1 mL of bacterial suspensions using a 1 mL needleless syringe. Lesion areas were measured at 6 dpi using a digital Vernier calliper.

### 2.5. Pathogenicity Test of the Isolated SA-Producing EB

To test whether the SA-producing EB-44 and EB-47 isolates were pathogenic to *N. benthamiana* plants, the isolates were infiltrated into the leaves of *N. benthamiana* at a concentration of OD_600_ 0.1. *Pst* and 10 mM MgCl_2_ were used as positive and negative controls, respectively. The incidence of wildfire disease symptoms was assessed up to 5 dpi.

### 2.6. Reisolation and Confirmation of SA-Producing EB

Reisolation of SA-producing EB isolates was performed to determine whether the EB could colonize the interior of *N. benthamiana* plants and affect their growth. *Nicotiana benthamiana* seeds were soaked in a suspension containing the SA-producing EB-44 at OD_600_ = 0.5 for 3 h and then grown for 4 weeks; the EB were subsequently reisolated from the leaves as described in the previous section. *Nicotiana benthamiana* seeds soaked in 10 mM MgCl_2_ were used as a control for comparison of plant growth. To confirm that the reisolated EB were the same as the original SA-producing isolate, the 16S rRNA gene sequences of the original and reisolated EB were compared. The experiment was repeated twice, with four replicates.

### 2.7. Quantification of Nicotine production in Tobacco Species Co-Cultivated with EB

Approximately 0.5 g of leaves collected from the upper parts of 4-week-old *N. benthamiana* and 8-week-old *N. tabacum* plants were ground in liquid nitrogen using a mortar and pestle and mixed with 3.5 mL of 90% methanol. The ground sample was sonicated in a sonicating water bath for 10 min to lyse the cells, and was then incubated for 3 h at room temperature at 100 rpm. The supernatant was collected in Eppendorf tubes, centrifuged at 14,000 × *g* for 10 min at 4 °C, and filtered using a 0.22 µm nylon filter for LC–MS analysis.

The filtered samples were transferred to autosampler vials and 5 µL aliquots were analysed by LC–MS. Nicotine levels were quantified using a LC–MS system composed of an HPLC apparatus (Model 2695; Waters, Milford, MA, USA) equipped with a reversed phase column (Luna C18, 4.6 × 150 mm, 5 μm; Phenomenex, Torrance, CA, USA) and a mass spectrometer (Model 3100; Waters). For HPLC, the conditions were as follows: the injection volume was 5 µL; the mobile phase consisted of 20 mM ammonium acetate buffer (solvent A, pH 7.2) and acetonitrile (solvent B); the gradient elution program was 50:50 (A/B) from 0 to 5 min, and a linear increase of B to 100% to 7 min, before returning to 50:50 (A/B) at 10 min at a flow rate of 0.5 mL/min. For the MS, the desolvation gas (N_2_) flow rate was 4 L/h, desolvation temperature was 350 °C, capillary voltage was 4 kV, cone voltage was 30 V, ionization mode was set to electrospray positive, and single ion recording was set at *m/z* = 163.

### 2.8. Statistical Analysis

All data are expressed as means ± standard deviations (SDs) of three or four independent replications from each experiment. Statistical analysis of the results was conducted using one-way analysis of variance (ANOVA) followed by Duncan’s multiple range test at *p* < 0.05 using the Statistical Analysis Software (SAS) (Version: SAS 9.4; SAS Institute Inc., Cary, NC, USA).

## 3. Results

### 3.1. Isolation of EB and Measurement of SA in EB and the Host Plants

In total, 134 EB were isolated from various tissues of the six selected plant species; namely: *Metasequoia glyptostroboides*, *Ginkgo biloba*, *Taxus brevifolia*, *Pinus densiflora*, *Salix babylonica*, and *S. chaenomeloides*. *Ginkgo biloba* and *M. glyptostroboides* were selected because both species are considered living fossil plants. Ginkgos are an ancient plant line, with the earliest representatives having been found in approximately 280-million-year-old rocks from the Permian age. Metasequoia glyptostroboides was first identified as a living fossil species in 1941, and its unmineralized stumps and leaves from the warm Eocene Epoch approximately 45 million years ago can still be observed. *Taxus brevifolia* and *P. densiflora* were selected as gymnosperms with medicinal uses, while *Salix babylonica* and *S. chaenomeloides* were selected as angiosperms and based on their accumulation of SA in their bark.

The SA concentrations in the leaves of *M. glyptostroboides*, *G. biloba*, *T. brevifolia*, *P. densiflora*, *S. babylonica*, *S. chaenomeloides*, and *N. benthamiana* were analysed. The leaves of *S. babylonica* and *N. benthamiana* presented SA levels of 966.50 µg/g and 833.67 µg/g of fresh weight (FW), respectively. The SA concentrations in the leaves of *M. glyptostroboides*, *G. biloba*, *T. brevifolia*, *P. densiflora*, and *S. chaenomeloides* were below the detectable limit. Endophytic bacteria were isolated from leaves (73 isolates, 54.48%), fruits (49 isolates, 36.57%), seeds (eight isolates, 5.97%), and cones (four isolates, 2.98%) (Table 1). Tissue-specific EB colony-forming units (CFUs) per gram of fresh tissue were in the range of 1.8 × 10^2^ to 6.0 × 10^4^ CFUs/g in fruits and 1.8 × 10^3^ to 1.5 × 10^5^ CFUs/g in leaves, and presented values of 7.2 × 10^3^ CFUs/g in seeds and 1.2 × 10^3^ CFUs/g in cones (Table 1).

When EB were identified by HPLC as producing SA, the peaks of the EB extracts exactly matched the retention time of the SA standard at 6.9 min. Of the 134 EB isolates, 27 could produce SA, and the highest percentage of SA-producing EB was obtained from *S. babylonica* (74.07%) (Table 1). When the SA-producing EB were identified by 16S rRNA gene sequencing (Appendix A), the two EB producing the greatest quantities of SA were *Pseudomonas tremae* (EB-44, 57.05 µg/mL) and *Curtobacterium herbarum* (EB-47, 46.22 µg/mL) (Figure 1 and Appendix A). The constructed phylogenetic tree indicated that the 16S rRNA sequences of EB-44 and EB-47 showed the greatest similarities to those of *P. tremae* TO1 and *C. herbarum* P420/07, respectively. These two SA-producing isolates (EB-44 and EB-47), along with one non-SA-producing isolate (EB-52; *C. plantarum*) as a control, were selected for further experiments. EB-44, EB-47, and EB-52 were isolated from the leaves of *S. babylonica*.

Pathogenicity tests were conducted to confirm the non-toxicity and non-pathogenicity of EB-44, EB-47, and EB-52 on *N. benthamiana.* In contrast to the brown lesions induced by *Pst* infiltration, no visible toxicity or disease symptoms were elicited in *N. benthamiana* leaves in response to EB infiltration.

### 3.2. Disease Resistance in Nicotiana plants Grown from Seeds Treated with SA-Producing EB

Resistance against *Pst*-induced disease in *N. benthamiana* was investigated in plants grown in a walk-in chamber from seeds soaked in an SA-producing EB suspension. Seeds of *N. benthamiana* were treated for 3 h with EB-44, EB-47, EB-52, or 10 mM MgCl_2_, and then grown for four weeks. With *Pst* infiltration into the leaves, the disease lesion area was considerably smaller in *N. benthamiana* plants grown from seeds treated with SA-producing EB than in those soaked in 10 mM MgCl_2_ or non-SA-producing EB (Figure 2). Control plants showed the largest lesion areas at 6 days post inoculation (dpi) (46.49 ± 1.93 mm), followed by plants treated with EB-52 (40.48 ± 5.54 mm), EB-47 (24.89 ± 2.99 mm), and EB-44 (15.23 ± 2.57 mm) (Figure 2a,b). The leaves of *N. benthamiana* grown from seeds treated with SA-producing EB-44 accumulated considerably higher SA levels (2823.17 µg/g FW) than those treated with 10 mM MgCl_2_ (1218.17 µg/g FW) (Figure 2c).

Resistance to *Pst*-induced disease in *N. tabacum* was further investigated in plants grown in a glasshouse from seeds treated for 3 h with SA-producing EB. Similar to that observed for the walk-in chamber, *N. tabacum* plants infiltrated with *Pst* and grown under glasshouse conditions presented markedly smaller lesion areas when treated with SA-producing EB than when treated with 10 mM MgCl_2_ or non-SA-producing EB (Figure 3). The greatest lesion area diameter at 6 dpi was observed in the control plants (29.24 ± 7.45 mm), followed by those treated with EB-52 (25.57 ± 5.21 mm), EB-47 (19.04 ± 1.06 mm), and EB-44 (15.06 ± 2.52 mm) (Figure 3a,b).

### 3.3. Growth Promotion of Nicotiana Plants Cultivated from Seeds Treated with SA-Producing EB

We evaluated the effect of SA-producing EB on the growth of *N. benthamiana* and *N. tabacum* grown from seeds treated with EB-44, EB-47, or EB-52 for 3 h. Compared to treatment with distilled water, seed EB inoculation increased both plant height and fresh weight. Under the growth chamber conditions used, *N. benthamiana* seed inoculation with EB-44, EB-47, or EB-52 increased plant height by 201.7%, 168.7%, and 160.9%, and increased fresh weight by 262.8%, 183.1%, and 173.6%, respectively, compared to distilled water treatment (Figure 4a–c). Under the greenhouse conditions, inoculation of *N. tabacum* seeds with EB-44, EB-47, or EB-52 increased plant height by 145.42%, 123.27%, and 105.34% and number of leaves by 124.32%, 116.22%, and 108.11%, respectively, compared to the control treatment (Figure 4d–f). For the two *Nicotiana* species and under both growth chamber and greenhouse conditions, two SA-producing isolates, EB-44 and EB-47, exerted greater plant growth-promoting effects than the non-SA-producing EB-52 isolate. As a positive control for verification of the effect of SA, 1 mM SA was sprayed on 7-week-old *N. tabacum* plants, which also resulted in increased plant height and leaf number by 103.31% and 105.41%, respectively (Figure 4d–f).

We used 16S rRNA sequencing to confirm that the EB isolated from *N. benthamiana* were the same SA-producing EB used in the original seed inoculation. Bacteria were isolated from the leaves of 4-week-old *N. benthamiana* plants co-cultured with EB-44, and their population densities were counted (1.64 × 10^5^ CFUs/g of fresh tissue). The bacteria reisolated from the leaves of plants grown from the EB-44-inoculated seeds were identified as EB-44 based on 16S rRNA gene sequences, and the 16S rRNA sequences of the isolates exactly matched that of the EB-44 used for the original seed inoculation.

### 3.4. Nicotine Accumulation in Nicotiana Inoculated with SA-Producing EB

The nicotine content in the leaves of *Nicotiana* species was analysed by liquid chromatography–mass spectrometry (LC–MS). Compared to the chromatogram and mass spectra of a nicotine standard, the extract of *N. benthamiana* grown from seeds inoculated with SA-producing EB showed a peak at the same retention time (5.04 min) and mass spectrum profile of the standard at 162.9 *m*/*z* (Figure 5a,b). The leaves of *N. benthamiana* grown from seeds inoculated with EB-44, EB-47, or EB-52 accumulated 4.6, 3.2, and 1.2-fold higher nicotine levels than those imbibed with a 10 mM MgCl_2_ solution, respectively.

Further experiments using SA-producing EB infiltration into leaves were performed to determine the direct infiltration of SA-producing EB into the leaves. Leaves infiltrated with EB-44, EB-47, or EB-52 contained 3.6, 2.9, and 1.4-fold higher nicotine concentrations, respectively, than those imbibed with a 10 mM MgCl_2_ solution (Figure 5c,d).

All leaf extracts from *N. tabacum* grown from seeds treated with SA-producing EB, non-SA-producing EB, or 10 mM MgCl_2_ exhibited LC peaks at 4.49 min (Figure 6a), exactly matching the retention time of the nicotine standard. Under glasshouse growth conditions, leaf nicotine concentrations increased 1.4, 1.3, 1.2, and 1.1-fold in EB-44, EB-47, EB-52, and SA-treated plants, respectively (Figure 6b).

## 4. Discussion

We aimed to determine the effects of SA-producing EB on the resistance of two host tobacco species against tobacco wildfire disease, one of the most devastating diseases affecting tobacco and for which no effective control measures currently exist. Endophytic bacteria are known as bioprospecting microorganisms owing to their ability to synthesize novel bioactive compounds that can be used by plants for defence against pathogens [40]. In total, 134 EB isolates were obtained from the leaves, fruits, seeds, and cones of six plant species; namely: *M. glyptostroboides, G. biloba, T. brevifolia, P. densiflora, S. babylonica*, and *S. chaenomeloides*. The bacterial population densities, shown as CFUs per gram of fresh weight, varied depending on the plant species and tissue type (Table 1). Variations in bacterial population densities have been reported to depend on plant species, tissue type, location, and environmental conditions [41,42,43]. In our studied plant species, the highest EB density was found in leaves, except for *G. biloba*, where the EB density in the fruits was 33-fold higher than that in the leaves (Table 1).

Analysis by HPLC confirmed that 27 EB isolates could produce SA, with EB-44 producing the highest quantity (Figure 1 and Appendix A). In bacteria, SA is biosynthesized from chorismate via two reactions catalysed by isochorismate synthase and isochorismate pyruvate lyase [44]. We also quantified SA levels in the EB host plants to determine whether they accumulated SA. Among the host plants, *S. babylonica* produced the most SA (966.50 µg/g FW). The number of SA-producing EB isolates was strongly correlated with SA contents in the leaves of the host plants.

*Nicotiana benthamiana* plants grown from seeds soaked in SA-producing EB-44 and EB-47 accumulated higher levels of SA than those treated with non-SA producing EB-52 or control plants (Figure 2). Natural synthesis of SA in *N. tabacum* and *Solanum tuberosum* has been reported to be less than 100 ng and 10 µg/g fresh weight, respectively [45,46,47,48]. *Nicotiana tabacum* plants treated with SA-producing *S. marcescens* 90-166 accumulated markedly higher levels of SA [10], which may be related to increased resistance against pathogen-induced diseases [49].

Endophytic bacteria are known for their advantageous relationships with host plants and antagonistic activities against plant pathogens [50]. In our plant disease assays, inoculation of *N. benthamiana* and *N. tabacum* with EB-44 elicited a greater reduction in *Pst*-induced disease symptoms than the control treatment with 10 mM MgCl_2_ (Figure 2 and Figure 3). Reduced *Botrytis cinerea-*induced necrosis has been observed in beans in the presence of SA-producing *P. aeruginosa* 7NSK2 [9,51], while *N. tabacum* plants show significant resistance against tobacco necrosis virus when inoculated with the SA-producing *P. fluorescens* CHA0 strain [9,51]. Endophytic bacteria mediate increased host resistance to pathogen-induced disease in association with pathogen-induced SAR [52]. SA is converted to methyl salicylate by the action of SA carboxyl methyltransferase, which acts as an important long-distance signal mediating SAR [53,54].

In addition to the effect on SA accumulation, EB-44 and EB-47 treatment increased height, weight, and leaf number in the *N. benthamiana* and *N. tabacum* hosts (Figure 4). These effects could be mediated either by SA produced by the EB or by a growth-promoting effect of the EB themselves. In our study, exogenous SA application also exerted a positive effect on the growth of *N. tabacum* plants (Figure 4), indicating that growth promotion may be mediated by the higher levels of SA generated by SA-producing EB. Several studies have reported that exogenous SA application increases the growth of soybeans, *Arabidopsis*, and chamomile [55,56,57]. However, the effect of SA on plant growth is concentration-dependent, and at high concentrations, SA can also inhibit plant growth [57].

Nicotine is a naturally occurring pyridine alkaloid in *Nicotiana* species [24,58,59]. In addition to increasing the disease resistance in *N. benthamiana* and *N. tabacum*, EB-44 also induced the highest nicotine concentrations of all the treatments (Figure 5 and Figure 6). To the best of our knowledge, this is the first report on enhanced nicotine accumulation induced by SA-producing EB. Tobacco plants inoculated with *Pst* and *P. syringae* pv. *tomato* and infected with tobacco mosaic virus displayed increased nicotine accumulation [27,28,29]. Infection with *Pst* leads to the production of pipecolic acid in tobacco, triggering SA biosynthesis and increased nicotine accumulation [29].

Salicylic acid is widely used as an elicitor to stimulate the synthesis of secondary metabolites [60]. The accumulation of nicotine and other related alkaloids in tobacco can also be affected by environmental factors and levels of plant hormones, including that of JA [61,62,63]. Nicotine accumulation was shown to be inhibited by SA application, whereas it was enhanced by wounding in tobacco plants [64]. In contrast, in our study, we found that SA application induced nicotine accumulation, possibly because nicotine synthesis in tobacco plants is regulated by multiple signalling molecules, including JA and/or auxin [30], and a high SA concentration has been reported to inhibit JA signal transduction [65]. Alternatively, SA-producing EB might induce the production of some amino acids in tobacco plants that lead to increased nicotine and SA synthesis.

In conclusion, our study reports novel roles for EB, including in SA production; promotion of SA biosynthesis and disease resistance in inoculated plants; and production and increased accumulation of nicotine in tobacco. The SA-producing isolate EB-44 (*P. tremae*) was the most effective at suppressing tobacco wildfire disease and has potential for use as an alternative, eco-friendly control measure for this disease. Our results indicated that *P. tremae* showed the most promise in this role, possibly because it contains antibacterial compounds. To the best of our knowledge, the increased accumulation of nicotine in tobacco plants observed in this study is a new EB-associated phenomenon. Further studies are needed to reveal the mechanism underlying EB-mediated enhancement of nicotine accumulation in tobacco; in addition, the antibacterial compounds produced by this EB should also be identified for further formulation and future applications in the field.

## Figures and Tables

**Figure 1 microorganisms-08-00031-f001:**
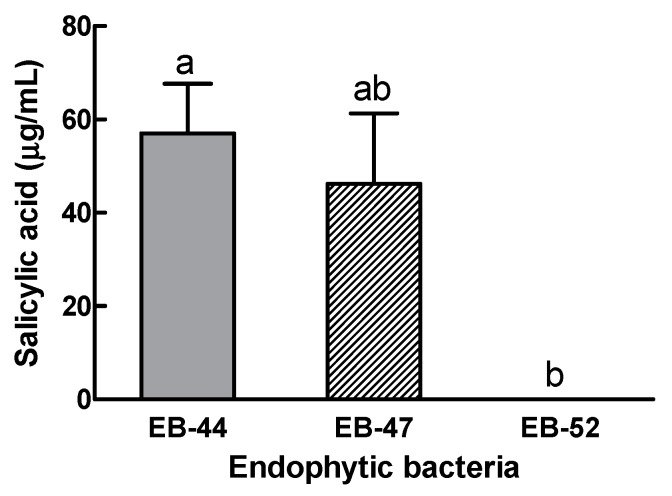
Salicylic acid concentrations in EB-44 (*Pseudomonas tremae*), EB-47 (*Curtobacterium herbarum*), and EB-52 (*C. plantarum*) isolated from the leaves of *Salix babylonica*. Data represent means ± standard deviations of four independent replications for each treatment. Means with different letters are significantly different at *p* < 0.05 by Duncan’s multiple range test.

**Figure 2 microorganisms-08-00031-f002:**
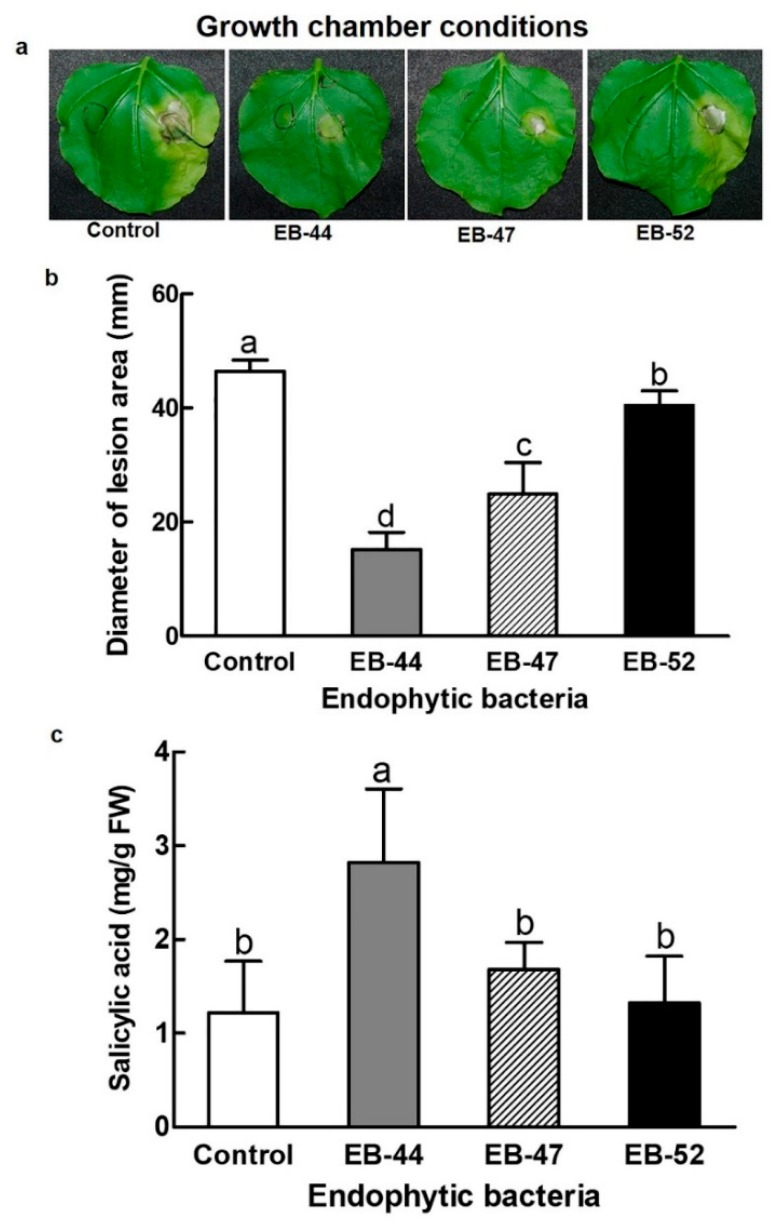
Disease resistance against *Pseudomonas syringae* pv. *tabaci* (*Pst*) and salicylic acid (SA) concentrations in the leaves of *Nicotiana benthamiana* grown from seeds treated with 10 mM MgCl_2_ (control), EB-44, EB-47, or EB-52 for 3 h. (**a**,**b**) The leaves of 4-week-old plants were infiltrated with 0.1 mL of *Pst* and the diameters of the infected areas were measured 6 days post inoculation. Left: 10 mM MgCl_2_ as control; right: 0.1 OD_600_
*Pst*. (**c**) The concentrations of SA in the leaves were measured before *Pst* infiltration. Data represent means ± standard deviations of four independent replications for each treatment. Means with different letters are significantly different at *p* < 0.05 by Duncan’s multiple range test. FW = fresh weight.

**Figure 3 microorganisms-08-00031-f003:**
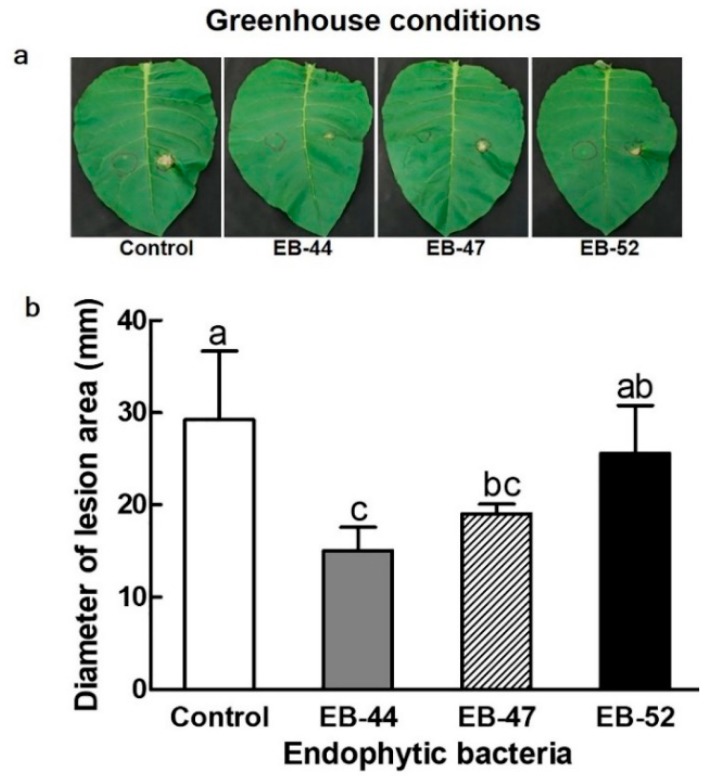
The effect of salicylic acid-producing endophytic bacteria on the resistance of *Nicotiana tabacum* against *Pseudomonas syringae* pv. *tabaci (Pst). Nicotiana tabacum* plants grown from seeds treated with 10 mM MgCl_2_ (control), EB-44, EB-47, or EB-52 for 3 h were evaluated for disease incidence in 4-week-old leaves infiltrated with 0.1 mL of *Pst*; infection diameter was measured 6 days after inoculation. (**a**) Left: 10 mM MgCl_2_ treatment as control; right: 0.1 OD_600 _*Pst*. (**b**) Lesion diameter. Data represent means ± standard deviations of four independent replications for each treatment. Means with different letters are significantly different at *p* < 0.05 by Duncan’s multiple range test.

**Figure 4 microorganisms-08-00031-f004:**
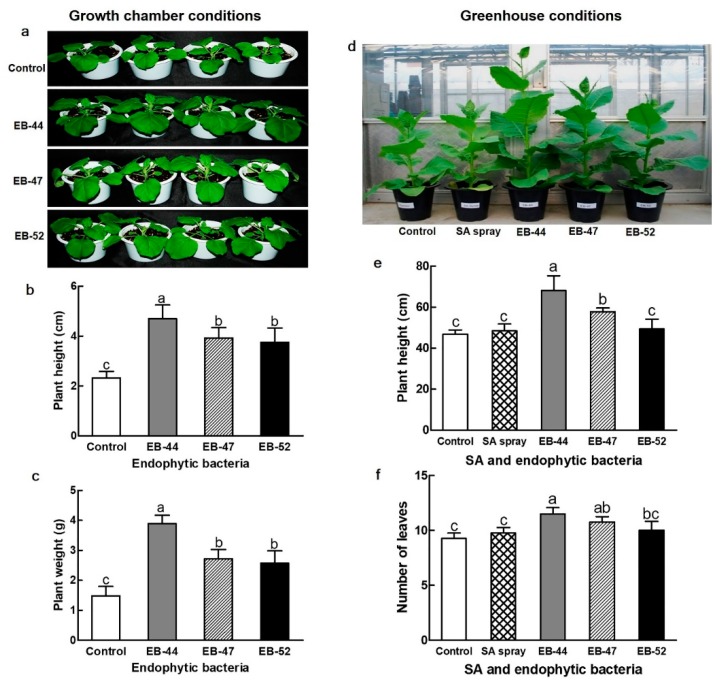
Growth-promoting effect of salicylic acid (SA)-producing endophytic bacteria (EB) in *Nicotiana.* (**a**–**c**) Growth comparison of 3-week-old *Nicotiana benthamiana* plants in response to seed treatment with SA-producing EB or distilled water (Control). (**d**–**f**) Growth comparison of 8-week-old *N. tabacum* plants in response to seed treatment with SA-producing EB or water (Control). Data represent means ± standard deviations of four independent replications for each treatment. Means with different letters are significantly different at *p* < 0.05 by Duncan’s multiple range test.

**Figure 5 microorganisms-08-00031-f005:**
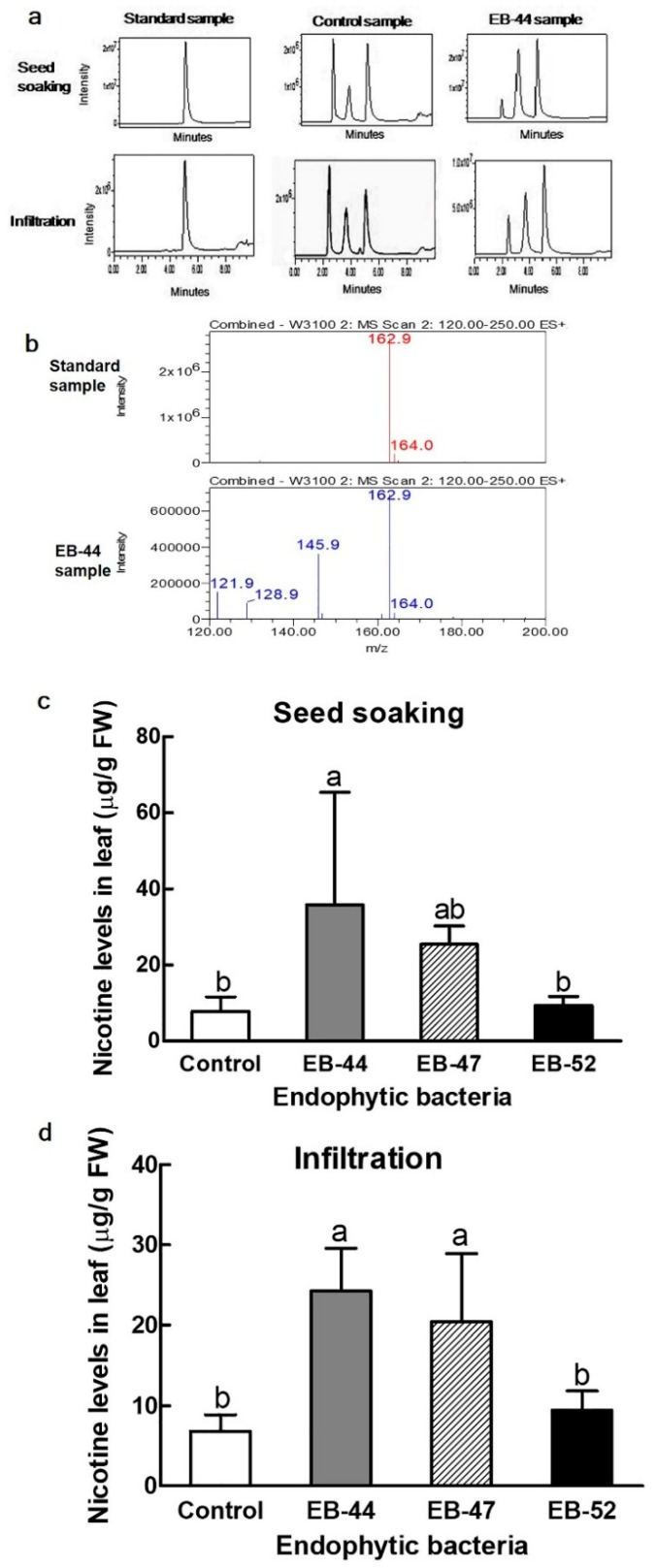
Liquid chromatography–mass spectrometry (LC–MS) analysis for the identification and quantification of nicotine levels in the leaves of *Nicotiana benthamiana* treated with salicylic acid (SA)-producing endophytic bacteria (EB). (**a**) LC–MS chromatograms of the standard, control, and EB-44 samples with matching nicotine peaks at 5.04 min. (**b**) Mass spectra profiles of the matching (162.9 *m*/*z*) nicotine standard and EB-44. (**c**,**d**) Nicotine contents in the leaves of *N. benthamiana* grown from seeds treated with SA-producing EB or by leaf infiltration. Data represent means ± standard deviations of four independent replications for each treatment. Means with different letters are significantly different at *p* < 0.05 by Duncan’s multiple range test. FW = fresh weight.

**Figure 6 microorganisms-08-00031-f006:**
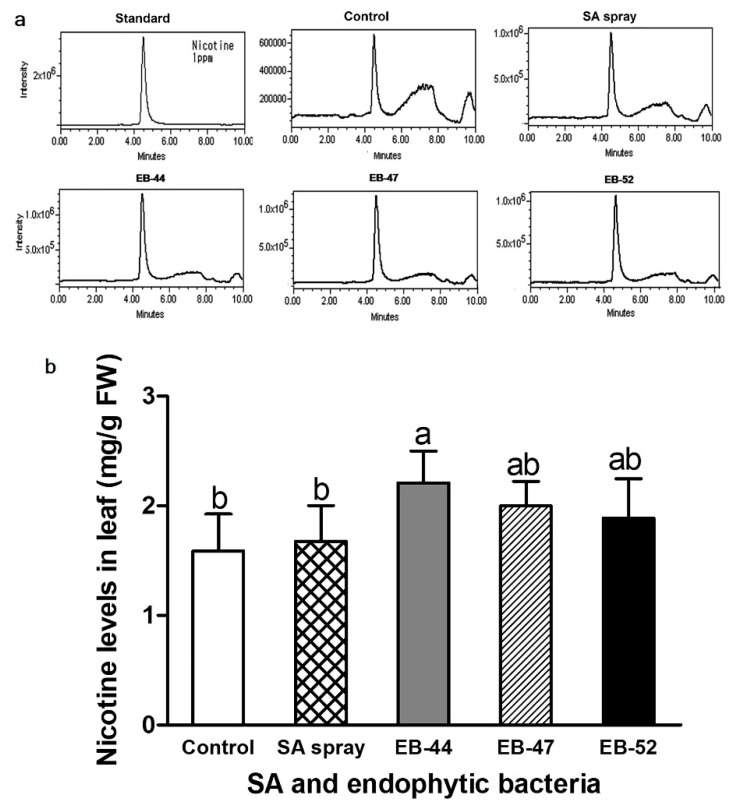
Liquid chromatography–mass spectrometry (LC–MS) analysis for the identification and quantification of nicotine levels in the leaves of *Nicotiana tabacum* treated with salicylic acid (SA)-producing endophytic bacteria (EB) or SA sprayed on plant leaves. (**a**) Completely matched LC–MS chromatograms of the standard sample with the nicotine peak at 4.49 min and the EB-44 sample with the peak at 4.49. (**b**) Nicotine content in the leaves of *N. tabacum.* Data represent means ± standard deviations of four independent replications for each treatment. Means with different letters are significantly different at *p* < 0.05 by Duncan’s multiple range test. FW = fresh weight.

**Table 1 microorganisms-08-00031-t001:** Isolation of salicylic acid-producing endophytic bacteria from six plant species.

Plant Species	Total no. of Isolates	Tissue	No. of Isolates	CFUs/g of Fresh Tissue	No. of EB Isolates	
					SA	No SA
*Metasequoia* *glyptostroboides*	8	Leaves	4	3.3 × 10^4^	0	4
		Cones	4	1.2 × 10^3^	2	2
*Ginkgo biloba*	50	Leaves	11	1.8 × 10^3^	1	10
		Fruits	39	6.0 × 10^4^	1	38
*Taxus brevifolia*		Leaves	23	8.6 × 10^4^	2	21
	41	Seeds	8	7.2 × 10^3^	0	8
		Fruits	10	1.8 × 10^2^	1	9
*Pinus densiflora*	2	Leaves	2	9.6 × 10^3^	0	2
*Salix babylonica*	24	Leaves	24	1.5 × 10^5^	20	4
*Salix chaenomeloides*	9	Leaves	9	5.9 × 10^4^	0	9
Total	134		134	-	27	107

EB: endophytic bacteria; SA: salicylic acid; CFUs = colony forming units.

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
