# Peer review of "Salicylic Acid-Producing Endophytic Bacteria Increase Nicotine Accumulation and Resistance against Wildfire Disease in Tobacco Plants"

_microorganisms, 2019, doi:10.3390/microorganisms8010031_

Round 1

Reviewer 1 Report

Dear Authors

I read this manuscript entitled “Salicylic Acid-Producing Endophytic Bacteria Increase Nicotine Accumulation and Resistance against Wildfire Disease in Tobacco plants (Manuscript ID: microorganisms-638947)” which is interesting for clear mechanism of plant-endophyte relationship. And it is good to observe SA concentration after artificial infection. But this journal is “Microorganisms”. I hope that you will focus microorganisms and report about them. I felt there were too few events related to pathogens and endophytes in this report. Therefore, I decided that it was not suitable for this journal. I recommend that you make changes and submit them to another natural product journal.

I will point out the parts that I have reviewed and found particularly important. I will write to improve this paper

About the title

In modern society, where the health damage of tobacco is serious, the increase in nicotine has a bad impression. Will you be able to report including nicotine decrease?

About the Materials and Method

Line 92

Looking at the supplement diagram, you identify the species. I think 16S rDNA alone is not enough for identification in species.

Line 103

-evaporated at 50oC on a rotary evaporator

I think it is too high. Can ethyl acetate be removed below 50 degrees?

Line 146

This experiment needs to be done for EB-47 and EB-52.

Line 149-151

-To confirm that……….with four replicates

I think 16S rDNA alone is not enough for identification to same with EB-44.

There is no proof that there is no same fungus in tobacco (N. benthamiana) endophyte.

About the results

Line 193 Table 1

There is an unnecessary line under the Taxus.

Line 198

Isn't this plant (Salix babylonica and Salix chaenomeloides) just enough?

What does it mean to include other plants in this report?

Line 200-201 Fig. 1 and Fig. 2 and Fig. 3

EB-50 also produces SA, but why isn't it here?

What kind of microorganism is EB-52 not on the list?

I think that you need to be listed all endophytic fungi associated with Salix sp.

Line 223-224

An activity test between wildfire microbes and SA, and an activity test between wildfire microbes and EB-44 extract are required.

Line 274 Fig. 4

The conditions here are different. It is unclear whether the effect is due to the growth conditions or due to the difference in species.

The a, b and c should have a SA spray test.

About the Discussion

Line 317

-the most devastating diseases affecting tobacco

If so, you should write a paper focusing on this pathogen.

Line 323 Table 1

Does the same experiment produce the same result?
You need to show the validity of this number, such as showing the number of attempts.

Line 331-332

Where is this description in the text? It is necessary to show an experiment or literature.

Line 359-373

This part and the result of this time are not consistent. It is necessary to convince the reader with simple sentences.

Line 379 and 382

Why are “antibacterial compounds” suddenly described?

Author Response

Author’s response to reviewers’ comments

We appreciate the reviewers’ critical comments. We prepared our answers for the comments from the three reviewers one-by-one. Our answers for the comments are as follows;

Dear Authors

I read this manuscript entitled “Salicylic Acid-Producing Endophytic Bacteria Increase Nicotine Accumulation and Resistance against Wildfire Disease in Tobacco plants (Manuscript ID: microorganisms-638947)” which is interesting for clear mechanism of plant-endophyte relationship. And it is good to observe SA concentration after artificial infection. But this journal is “Microorganisms”. I hope that you will focus microorganisms and report about them. I felt there were too few events related to pathogens and endophytes in this report. Therefore, I decided that it was not suitable for this journal. I recommend that you make changes and submit them to another natural product journal.

I will point out the parts that I have reviewed and found particularly important. I will write to improve this paper

About the title:

In modern society, where the health damage of tobacco is serious, the increase in nicotine has a bad impression. Will you be able to report including nicotine decrease?

 Response: We appreciate the reviewer’s comment. Tobacco is undeniably one of the most important and avoidable causes of premature death and disease in the world. If we can increase the nicotine amount in leaves, then consumers will smoke less amount of tobacco because of higher levels of nicotine inside. Consumers should smoke more amount of tobacco leaves with less concentrations of nicotine, however, if high concentration can be accumulated in the leaves, then smokers smoke less amount of tobacco leaves; better for health because tobacco plants contain lots of detrimental compounds.  

About the Materials and Method

Line 92:

Looking at the supplement diagram, you identify the species. I think 16S rDNA alone is not enough for identification in species.

 Response: We appreciate the reviewer for the kind suggestion. Actually, we have some unpublished data which fascinated us and depending upon these data, we are planning to sequence the whole genome and do RNA sequencing in order to get insights about those endophytes. Hope, in our next manuscript we will able to include those sequenced data and clearly identified the endophytes.

But usually people use this 16S rDNA, and also these bacterial spp. is not difficult to be determined by the 16S rDNA sequencing.

Shahzad et al. (2016) Seed-borne endophytic Bacillus amyloliquefaciens RWL-1 produces gibberellins and regulates endogenous phytohormones of Oryza sativa. Plant Physiology and Biochemistry, Volume 106, September 2016, Pages 236-243. Identified the endophytic bacteria by the 16S rRNA sequencing.

Zhang et al. (2015). Rhizobium oryzicola sp. nov., potential plantgrowth-promoting endophytic bacteria isolated from rice roots. International Journal of Systematic and Evolutionary Microbiology, 65, 2931–2936. Identified the endophytic bacteria by the 16S rRNA sequencing.

Lata et al. (2006) Identification of IAA-producing endophytic bacteria from micropropagated Echinacea plants using 16S rRNA sequencing. Plant Cell, Tissue and Organ Culture 85: 353–359. Identified the endophytic bacteria by the 16S rRNA sequencing and deposited in the GenBank database.

Khan & Doty (2009). Characterization of bacterial endophytes of sweet potato plants. Plant Soil 322:197–207. Identified the endophytic bacteria by the 16S rRNA sequencing.

Line 103:

-evaporated at 50oC on a rotary evaporator. I think it is too high. Can ethyl acetate be removed below 50 degrees?

 Response: Ethyl acetate can be removed below 50 degrees but the evaporation process will slow and it will take more time to evaporate ethyl acetate.

Line 146: This experiment needs to be done for EB-47 and EB-52.

Response: We appreciate the reviewer’s comment. The experiment can be done for EB-47 and EB-52. For more confirmation, we included two SA-producing EB (EB-44 & EB-47).

Line 149-151:

-To confirm that……….with four replicates

I think 16S rDNA alone is not enough for identification to same with EB-44.

There is no proof that there is no same fungus in tobacco (N. benthamiana) endophyte.

 Response: We appreciate the reviewer for the kind suggestion. Actually, we have some unpublished data which fascinated us and depending upon these data, we are planning to sequence the whole genome and do RNA sequencing in order to get insights about those endophytes. Hope, in our next manuscript we will able to include those sequenced data and clearly identified the endophytes.

But usually people use this 16S rDNA, and also these bacterial spp. is not difficult to be determined by the 16S rDNA sequencing.

Shahzad et al. (2016) Seed-borne endophytic Bacillus amyloliquefaciens RWL-1 produces gibberellins and regulates endogenous phytohormones of Oryza sativa. Plant Physiology and Biochemistry, Volume 106, September 2016, Pages 236-243. Identified the endophytic bacteria by the 16S rRNA sequencing.

Zhang et al. (2015). Rhizobium oryzicola sp. nov., potential plantgrowth-promoting endophytic bacteria isolated from rice roots. International Journal of Systematic and Evolutionary Microbiology, 65, 2931–2936. Identified the endophytic bacteria by the 16S rRNA sequencing.

Lata et al. (2006) Identification of IAA-producing endophytic bacteria from micropropagated Echinacea plants using 16S rRNA sequencing. Plant Cell, Tissue and Organ Culture 85: 353–359. Identified the endophytic bacteria by the 16S rRNA sequencing and deposited in the GenBank database.

Khan & Doty (2009). Characterization of bacterial endophytes of sweet potato plants. Plant Soil 322:197–207. Identified the endophytic bacteria by the 16S rRNA sequencing.

We did not deal with fungi for our experiments, we concentrated on the EB.

About the results

Line 193:

Table 1: There is an unnecessary line under the Taxus.

Response: As reviewer’s comment, we formatted the table and removed the line under the Taxus.

Line 198:

Isn't this plant (Salix babylonica and Salix chaenomeloides) just enough?

What does it mean to include other plants in this report?

 Response: We appreciate the reviewer’s comment. We included the data in our manuscript as we did our experiments. If we include only Salix babylonica and Salix chaenomeloides, it seems incomplete to us. As we did not get any comments from other two reviewer’s we want to keep all the data in the manuscript.

 Line 200-201:

Fig. 1 and Fig. 2 and Fig. 3.

EB-50 also produces SA, but why isn't it here?

What kind of microorganism is EB-52 not on the list?

I think that you need to be listed all endophytic fungi associated with Salix sp.

Response: We appreciate the reviewer’s critical comment. Among the 27 EB isolates that produced SA, the highest SA-producing strains, EB-44 and EB-47 were selected for experiments.

We also selected non-SA-Producing endophytic bacteria for our experiment as a control. EB-52 is Curtobacterium plantarum.

We added it intentionally because it did not produce SA but we’d like to check the effect of the EB on living in a host plant.

Line 223-224:

An activity test between wildfire microbes and SA, and an activity test between wildfire microbes and EB-44 extract are required.

 Response: We appreciate the reviewer’s suggestions. Due to time limitation and project completion report, it was could not those kind of experiments.

Line 274:

Fig. 4. The conditions here are different. It is unclear whether the effect is due to the growth conditions or due to the difference in species.

The a, b and c should have a SA spray test.

 Response: We have used two different species of tobacco. In each species, we have used separate plants for each treatment.

We conducted the SA spray test at green house for N. tabacum. We could not complete the SA spray test for N. benthamiana due to the space limitation of our in vitro growth chamber.

About the Discussion

Line 317:

-the most devastating diseases affecting tobacco

If so, you should write a paper focusing on this pathogen.

 Response: We appreciate the reviewer’s for critical comment. There are many research papers related to the tobacco wildfire disease and its causal organisms Pseudomonas syringae pv. Tabaci. On the way of our research, if we get something new, we will write paper focusing on this pathogen.

Line 323:

Table 1, Does the same experiment produce the same result?
You need to show the validity of this number, such as showing the number of attempts.

Response: We repeated the experiment three times and we got almost similar results. From those results we randomly selected one and included in our manuscript.

Line 331-332:

Where is this description in the text? It is necessary to show an experiment or literature.

Response: We appreciate the reviewer’s comment. In Line 331-332 of our manuscript we have written ‘We also quantified SA levels in the EB host plants to determine whether they accumulated SA. Among the host plants, S. babylonica produced the most SA (966.50 µg/g FW).”

We added some text related to this in the manuscript as “The SA concentrations in the leaves of M. glyptostroboides, G. biloba, T. brevifolia, P. densiflora, S. babylonica, S. chaenomeloides, and N. benthamiana were analysed. The leaves of S. babylonica and N. benthamiana presented SA levels of 966.50 µg/g and 833.67 µg/g of fresh weight (FW), respectively. The SA concentrations in the leaves of M. glyptostroboides, G. biloba, T. brevifolia, P. densiflora, and S. chaenomeloides were below the detectable limit.” (Line 187-191).

Line 359-373:

This part and the result of this time are not consistent. It is necessary to convince the reader with simple sentences.

Response: We appreciate the reviewer’s critical comment. Two figures (Fig. 5 and 6) are related to this discussion part. As another two reviewer’s did not comment on this, we do not want to change this part of discussion.

Line 379 and 382:

Why are “antibacterial compounds” suddenly described?

Response: We appreciate the reviewer’s comment. In line Line 379 and 382, we have described the future prospects of EB-44 and EB-47. So we have given emphasis on “antibacterial compounds” might produce by EB-44 and EB-47.

Reviewer 2 Report

The manuscript describes the resistance-inducing activity and growth-promoting activity by SA-producing endophytic bacteria isolated from Salix babylonica. These results provide useful information that the use of SA-producing endophytic bacteria is effective in disease protection and growth promotion of tobacco plants. However, there are some problems that needed to be addressed. My comments are listed below.

Line 70. "five plant species">>>"six plant species"

Line 108. "suspension of each EB" is not correct.

Line 118. "S. chaenomeloides" should be added.

Line 125, 146. "in a solution">>>"in a suspension"

Line 186. The SA concentrations in the leaves of M. glyptostroboides, G. biloba, T. brevifolia, P. densiflora and S. chaenomeloides should be described.

Line 200. The host plant of EB-44, EB-47 and EB-52 should be described in the text.

Line 207. The title of Fig. 1 should be written correctly.

Line 321. "five plant species">>>"six plant species"

Author Response

Author’s response to reviewers’ comments

We appreciate the reviewers’ critical comments. We prepared our answers for the comments from the three reviewers one-by-one. Our answers for the comments are as follows;

Comments and Suggestions for Authors

The manuscript describes the resistance-inducing activity and growth-promoting activity by SA-producing endophytic bacteria isolated from Salix babylonica. These results provide useful information that the use of SA-producing endophytic bacteria is effective in disease protection and growth promotion of tobacco plants. However, there are some problems that needed to be addressed. My comments are listed below.

Line 70:

"five plant species">>>"six plant species"

Response: We appreciate the reviewer’s critical comment. We changed the five plant species to six plants species in the manuscript (Line 176, 178, 321).

Line 108:

"suspension of each EB" is not correct.

Response: We replaced the word “suspension” by “solution”.

Line 118:

"S. chaenomeloides" should be added.

 Response: We added the S. chaenomeloides.

Line 125, 146:

"in a solution">>>"in a suspension"

Response: We changed the "in a solution" to "in a suspension"

Line 186:

The SA concentrations in the leaves of M. glyptostroboides, G. biloba, T. brevifolia, P. densiflora and S. chaenomeloides should be described.

Response: As per reviewer’s comment, We added a sentence “The SA concentrations in the leaves of M. glyptostroboides, G. biloba, T. brevifolia, P. densiflora, and S. chaenomeloides were below the detectable limit”.

Line 200:

The host plant of EB-44, EB-47 and EB-52 should be described in the text.

Response: As per reviewer’s comment, We added a sentence "The EB-44, EB-47 and EB-52 were isolated from the leaves of S. babylonica.

Line 207:

The title of Fig. 1 should be written correctly.

Response: We appreciate the reviewer’s comment. We changed the title of the figure as “Salicylic acid concentrations in EB-44 (P. tremae), EB-47 (C. herbarum), and EB-52 (C. plantarum) isolated from the leaves of S. babylonica. Data represent means ± standard deviation of four independent replications for each treatment. Means with different letters are significantly different at p < 0.05 by Duncan’s multiple range test.”

Line 321:

"five plant species">>>"six plant species"

 Response: We changed the five plant species to six plants species in the manuscript (Line 176, 178, 321).

Reviewer 3 Report

Nurul Islam and colleagues isolated bacteria from varies tree material and tested their potential against wildfire disease in tobacco plants. The article is well written and results are presented in a straight way. Findings about the reduction of infection rates by the bacteria are conclusive. As well as the SA leaf pattern after inoculation with the different bacteria.  

Two minor comments from my side. Do they authors know the study by Qin et al., 2019 `Responses of phyllosphere microbiota and plant health to application of two different biocontrol agents`? Qin found that Tobacco wildfire disease infection rates were significantly decreased by the foliar application of a bacterial mix of Bacillus (87.74%), Alcaligenes (7.69%), Pseudochrobactrum (2.86%) and Achromobacter (1.05%). I would recommend to add this finding to the introduction/discussion part.

I also wonder why the authors choose tree material as a potential source for bacterial antagonists of wildfire disease in tobacco. Is there any explanation?

Author Response

Author’s response to reviewers’ comments

We appreciate the reviewers’ critical comments. We prepared our answers for the comments from the three reviewers one-by-one. Our answers for the comments are as follows;

 Comments and Suggestions for Authors

Nurul Islam and colleagues isolated bacteria from varies tree material and tested their potential against wildfire disease in tobacco plants. The article is well written and results are presented in a straight way. Findings about the reduction of infection rates by the bacteria are conclusive. As well as the SA leaf pattern after inoculation with the different bacteria. 

Two minor comments from my side. Do they authors know the study by Qin et al., 2019 `Responses of phyllosphere microbiota and plant health to application of two different biocontrol agents`? Qin found that Tobacco wildfire disease infection rates were significantly decreased by the foliar application of a bacterial mix of Bacillus (87.74%), Alcaligenes (7.69%), Pseudochrobactrum (2.86%) and Achromobacter (1.05%). I would recommend to add this finding to the introduction/discussion part.

Response: We appreciate the reviewer for sincerely reviewing our manuscript. We included text along with reference as “Recent study shows that tobacco wildfire disease infection rates were significantly decreased by the foliar application of a bacterial mix of Bacillus (87.74%), Alcaligenes (7.69%), Pseudochrobactrum (2.86%) and Achromobacter (1.05%) [23] (Line 56-59)

Question: I also wonder why the authors choose tree material as a potential source for bacterial antagonists of wildfire disease in tobacco. Is there any explanation?

Response: We appreciate the reviewer’s comment. We have included some explanation in the manuscript. Ginkgo biloba and M. glyptostroboides were selected because both species are considered living fossil plants. Ginkgoes are an ancient plant line, with the earliest representatives having been found in approximately 280-million-year-old rocks from the Permian age. Metasequoia glyptostroboides was first identified as a living fossil species in 1941, and its unmineralized stumps and leaves from the warm Eocene Epoch approximately 45 million years ago can still be observed. Taxus brevifolia and P. densiflora were selected as gymnosperms with medicinal uses, while Salix babylonica and S. chaenomeloides were selected as angiosperms and based on their accumulation of SA in their bark. (Line 180-186).

Round 2

Reviewer 1 Report

Dear Author

I read this manuscript entitled “Salicylic Acid-Producing Endophytic Bacteria Increase Nicotine Accumulation and Resistance against Wildfire Disease in Tobacco plants (Manuscript ID: microorganisms-638947)”. which is interesting for clear mechanism of plant-endophyte relationship. And it is good to observe SA concentration after artificial infection. But this journal is “Microorganisms”. I hope that it will focus microorganisms and reported about them. I felt there were too few events about microorganisms like that related to pathogens and endophytes in this report. Therefore, I decided that it was not suitable for this journal. I recommend that you make changes and submit them to another natural product journal.

              I read revised version again, but the manuscript was not so better change for “Microorganisms”. I think that infected plant had another community of microbe in the plant. However, you have not mentioned about microbe in this report, reported only finding about “reduce pathogen circle” and “Nicotine increase”. Both of those are plant findings. So, for this journal lacks new information on microorganisms. In addition to, I couldn't understand your opinion about nicotine and health. This is very important in modern society and needs to be considered more deeply.

I did not write down the amendments to the details.

Thank you for listening to my opinion.

Sincerely yours